# The Influence of Human Factors Training in Air Rescue Service on Patient Safety in Hospitals: Results of an Online Survey

**DOI:** 10.3390/medicines10010002

**Published:** 2022-12-22

**Authors:** Christian von Rüden, Andre Ewers, Andreas Brand, Sven Hungerer, Christoph J. Erichsen, Philipp Dahlmann, Daniel Werner

**Affiliations:** 1Department of Trauma Surgery, BG Unfallklinik Murnau, 82418 Murnau, Germany; 2Institute for Biomechanics, BG Unfallklinik Murnau, 82418 Murnau, Germany; 3Institute for Biomechanics, Paracelsus Medical University, 5020 Salzburg, Austria; 4Institute of Nursing Science and Practice, Paracelsus Medical University, 5020 Salzburg, Austria; 5Center for Academic Further Education, Deggendorf Institute of Technology, 94469 Deggendorf, Germany; 6Department of Anesthesiology, LMU Klinikum, University Hospital, Ludwig-Maximilians-University, 80539 Munich, Germany; 7Department of Medicine, ADAC Air Rescue Service, 80686 Munich, Germany

**Keywords:** patient safety, crew resource management, human factors, team training, online survey, helicopter emergency medical services technical crew members

## Abstract

**Background**: Air rescue crew members work equally in aviation and medicine, and thus occupy an important interface between the two work environments of aviation and medicine. The aim of this study was to obtain responses from participants to a validated online-based questionnaire regarding whether hospitals may benefit from the commitment of a medical hospital staff which is also professionally involved in the aviation system as emergency physicians and Helicopter Emergency Medical Services Technical Crew Members (HEMS TC). Furthermore, it focused on the question of whether the skills acquired through Crew Resource Management (CRM) training in the air rescue service might also be used in the ground-based rescue service and, if so, whether they may have a positive effect. **Methods**: Medical air rescue staff of 37 German air rescue stations was included. Between 27 November 2020 and 03 March 2021, 253 out of 621 employees (response rate: 40.7%) participated voluntarily in a validated anonymized online survey. A quantitative test procedure was performed using the modified questionnaire on teamwork and patient safety (German version). **Results**: The examination and interpretation of the internal consistency (Cronbach’s alpha) resulted in the following reliabilities: Factor I (Cooperation): α = 0.707 (good); Factor II (Human factors): α = 0.853 (very good); Factor III (Communication): α = 0.657 (acceptable); and Factor IV (Safety): α = 0.620 (acceptable). Factor analysis explained 53.1% of the variance. **Conclusions**: The medical clinicians participating in this online survey believed that the skills they learned in human factors training such as CRM are helpful in their daily routine work in hospitals or other medical facilities, as well as in their ground-based rescue service activities. These findings may result in the recommendation to make CRM available on a regular to the medical staff in all medical facilities and also to ground-based rescue service staff aiming to increase patient safety and employee satisfaction.

## 1. Introduction

Patient safety has been in the public eye since the United States Institute of Medicine published the report “To err is human: Building a Safer Health System” in the year 2000 [1]. In Central Europe today, medical clinicians find themselves in an environment with a level of technology comparable to that of a modern aircraft. Standard operating procedures (SOPs) for operating complex technical systems and clear communication are tools to counter this challenge. However, if an unexpected event occurs, no SOP may help [2]. In air rescue service, this problem is addressed among others through Crew Resource Management (CRM) training [3], which focuses on preventing errors in human factors [4]. The training includes the parameters of cooperation, situational awareness, leadership, and decision making. The parameter communication is a permanent process, which can also be found in the other categories. Air rescue service organizations usually train every crew member in CRM and maintain the skills learned in regular refresher courses. Through CRM training, all available human and material resources should be used effectively for a safe and efficient overall process aiming to establish a comprehensive safety culture [5]. The transfer of essential elements of this safety culture from aviation to the working environment of medicine has led to a relevant improvement in patient safety and employee satisfaction [6]. Unfortunately, in medicine, it has long been assumed that interested individuals only would acquire interpersonal skills on their own [7].

Air rescue crew members work equally in aviation and medicine, and thus occupy an important interface between the two work environments of aviation and medicine [8]. In this respect, they seem particularly suited to assess whether the skills learned in air rescue also improve their everyday work in the hospital or another medical facility and make it safer, provided that they also use them consistently.

Nowadays, it is common consent that community paramedicine programs and trainings are diverse and also allow ground-based rescue service members to address a wide range of social needs [9]. On the other hand, these trainings have been relatively poorly described in the literature [10]. There is a lack of information as well as knowledge gaps on human factors training outcomes and on the necessity for the development of formalized education frameworks [11].

Patient safety as a central element in everyday hospital life has become largely objectifiable through research methods from organizational research. In this context, the aim of this study was to obtain responses from participants to a validated online-based questionnaire on teamwork and patient safety. It was of particular interest if hospitals might benefit from the commitment of a medical hospital staff (physicians and nurses) which is also professionally involved in the aviation system as emergency physicians and Helicopter Emergency Medical Services Technical Crew Members (HEMS TC). Furthermore, it focused on the question of whether the skills acquired through human factors trainings in the air rescue service might also be used in the ground-based rescue service and, if so, whether they might have a positive effect.

## 2. Methods

The medical air rescue staff of 37 air rescue stations managed by the ADAC Air Rescue Service gGmbH, Munich, Germany, was included. A total of 621 persons were working in their regular jobs as medical doctors or specialist nurses in a hospital or other medical facility and at the same time as helicopter emergency physicians or HEMS TC. In the period between 27 November 2020 and 03 March 2021, 253 out of these 621 employees (response rate: 40.7%) voluntarily participated in the validated anonymized online-based survey. A quantitative test procedure was used with the modified questionnaire on teamwork and patient safety (FTPS; Table 1), the German version of the English language Safety Attitudes Questionnaire (SAQ), which in turn was derived from the Flight Management Attitudes Questionnaire (FMAQ) [12,13,14]. The modified FTPS uses general items (Q1 to Q6) about education level, experience, and work situation followed by 33 items (Q7 to Q39) which are rated using a 5-point Likert scale (5 = strongly agree, 4 = agree, 3 = neutral, 2 = slightly disagree, and 1 = strongly disagree). The modified FTPS covers teamwork, safety climate, and perceptions of management. Items 7 to 12 are targeted questions about the effect of the use of CRM trainings. Items 13 to 24 assess teamwork climate, items 25 and 35 examine the safety climate, and items 36 to 39 evaluate respondents’ perceptions of hospital management and actions. The original FTPS provided by Salem et al. 2008 (items 13 to 39) was supplemented with items 1 to 12, adapted to the specific context of the current study. Neutral responses were assessed as indicating that the respondents were possibly cautious despite a rather negative attitude towards the statements, and therefore cast a neutral vote instead of a negative vote.

The cross-sectional online-based participation in this survey was carried out in accordance with all data protection guidelines by an anonymous link sent via the business e-mail address. All collected data were managed using SurveyMonkey^®^ (SurveyMonkey Europe UC, Dublin, Ireland). Five minutes were allotted for answering the questionnaire. Statistical data analysis was performed using SPSS version 19.0 (SPSS Inc., Chicago, IL, USA). Due to the multifactorial construction of the questionnaire, multidimensionality was to be expected. Therefore, factor analysis with Varimax rotation was used for questions 7 to 39 (33 items) for identification of testing scales (Table 2). Factor assignment using calculated eigenvalues was performed using the Kaiser criterion (eigenvalue ≥ 1) [15]. Verification of the internal validity of the identified testing scales was performed using Cronbach’s alpha.

## 3. Results

A total of 253 complete surveys were available for statistical analysis. The gender distribution was 79% men to 21% women. The modified FTPS took an average of 5 min and 3 s to complete. Using factor analysis and the Scree test, the extraction of four or five factors was proposed, whereby content considerations led to the decision of a four-factor solution [16]. The eigenvalue progression underlying the Scree test is demonstrated in Figure 1.

Factor analysis based on the Kaiser criterion revealed six factors, while logical assignment of items and factor reduction resulted in the allocation of the individual statements to the four test scales (factors) as follows:-Scale I (“Cooperation”): Q13, Q14, Q15, Q16, Q17, Q18, Q19, Q20, Q25, Q26, Q28, Q34;-Scale II (“Human factors”): Q7, Q8, Q9, Q10, Q11, Q12, Q23;-Scale III (“Communication”): Q21, Q22, Q24, Q29, Q31, Q32, Q33, Q35;-Scale IV (“Safety”): Q27, Q30, Q36, Q37, Q38, Q39.

The examination and interpretation of the internal consistency (Cronbach’s alpha) according to the instructions provided by Streiner 2003 resulted in the following reliabilities: Factor I: α = 0.707 (good); Factor II: α = 0.853 (very good); Factor III: α = 0.657 (acceptable); and Factor IV: α = 0.620 (acceptable) [17].

The detailed results of the individual items of the questionnaire are summarized as follows: Results of items Q1 to Q6 are presented in Table 3 and results of items Q13 to Q39 in Figure 2. The vast majority of over 90% of respondents agreed that the skills they learned in air rescue through CRM or similar training, such as communication skills or situational awareness, were also applicable in the hospital setting and that they had a positive impact on their ancestral occupation there. The same amount of persons agreed that human factors training and the skills that can be learned in it should be made available to all hospital staff involved in patient care. In total, 80% of respondents felt that applying air rescue skills to their everyday activities in the hospital made them feel more confident and satisfied than without such a background. Nearly the same number of persons agreed that applying skills learned in air rescue to everyday hospital life can improve collaboration between physicians, nurses, and other professionals. The statement that transferring skills and structures used and experienced in air rescue can improve patient safety in the hospital also received 80% agreement. Over 90% of the study participants who also work in ground-based ambulance services confirmed that the skills they learned in air-based ambulance services were applicable to and had a positive impact on their ground-based work.

## 4. Discussion

Behavioral concepts from aviation that can be applied in everyday situations are made available to some employees in the healthcare sector, such as air rescue service staff [18]. In this context, non-technical skills such as human factors play a decisive role in the safety culture of both working environments in terms of preventing errors and undesirable events [19,20,21]. The transfer of essential elements of the aviation safety culture such as CRM training to the medical sector has been demonstrated to increase patient safety in hospitals [22]. Despite the costs of CRM training, it presents a financially viable way to systematically organize for quality improvement due to cost savings based on the reduction in avoidable adverse events [23]. In this context, the purpose of this study was to determine whether patient safety and employee satisfaction in hospitals may benefit from the engagement of highly trained medical professionals working in both air rescue service and medical facilities. To our knowledge, this was the first study on this topic in the literature.

First of all, 253 out of 621 participants voluntarily took part in the questionnaire, representing a response rate of 40.7%. One might speculate that only persons who found the questionnaire interesting had the intention to answer it. On the other hand, we can only speculate what persons who decided not to participate in the survey thought.

Second, most of the participants of this survey were at an advanced stage, not only in terms of age but also in terms of length of service in air ambulances and in medical facilities, suggesting that the respondents had relevant work experience in both fields. It was also noteworthy that about 85% of the respondents were in the medical service and only 15% were in the nursing service. Conversely, this means that only 15% of the HEMS TC participated in the survey. A primary analysis has not yielded a sufficient explanation of the cause and revealed that the survey reached all HEMS TC via the email distribution list, which at least could rule out a technical cause. Third, in the survey group of physicians, the naturally high number of anesthesiologists was noticeable, which reflected the actual reality of medical staffing in German air rescue organizations across all providers. Nevertheless, the aim of future studies should be to clarify whether significant differences exist with regard to the results between the surveyed service groups or the various medical specialties. In addition, a country comparison based on the identical survey of the employees of the sister organizations of the same air rescue service would be very interesting to clarify potential country-related peculiarities or differences.

Another aspect was that the factor analysis within the scope of this study revealed the four dimensions “Cooperation”, “Human factors”, “Communication”, and “Safety” to be satisfactory and, for the dimension “Human factors”, to even have very good variance explanation. In the eigenvalue progression, these four factors explained 53.1% of the variance, and thus more than in the pilot study for the validation of the questionnaire provided by Salem et al. (2008) with 46.9%. Based on these four dimensions, the statements of the questionnaire were assigned to four scales, all of which demonstrated sufficient internal consistency in each case. Therefore, the modified FTPS could serve as a reliable questionnaire on the influence of human factors training in air rescue service on patient safety in hospitals.

Ninety percent of the respondents in this study agreed that skills such as communication or situational awareness that they learned in air rescue through CRM are applicable to their daily work in hospitals and that they have a positive impact on their day-to-day professional activities there. These findings were confirmed by previous studies highlighting the importance of CRM to improve patient safety in medical settings and led to the recommendation to make established team trainings available by default to all medical professionals [19,24,25,26,27].

Furthermore, 80% of the respondents felt that applying skills from air ambulance trainings made them feel more confident and satisfied in their day-to-day activities in the hospital than without such a background. It was confirmed that employee satisfaction also improved in the course of the skills learned. Corresponding indications have already been found in previous studies, but no larger study group has yet been confronted with this statement [7,28,29]. Thus, the current research was the first one to clearly demonstrate that the skills learned and used in air rescue service have a direct positive effect on the respondents’ traditional occupational activities in the hospital or other medical facility.

The statement that patient safety in hospitals can be improved by transferring skills and structures used and experienced in air rescue service was agreed to by 80% of the study participants. In this respect, the current study confirmed the rare literature [30,31,32]. In addition, nearly 80% of respondents in this large collective agreed that applying skills learned in air rescue service to everyday hospital practice could improve collaboration among physicians, nurses, and other professionals [33]. Therefore, the results of this study may encourage the operators of air rescue companies and the sponsors of medical facilities to enter into cooperation in the sense of a win–win situation through which they mutually benefit from the quality of the excellently trained airborne medical staff. In this connection, it might be recommended to locate air rescue stations at suitable hospitals.

The vast majority of the study participants felt that human factors training should be made available to all hospital staff working in patient care. While previous studies have addressed this issue and confirmed the benefits of human factors training in the healthcare sector, no previous study confronted such a highly specialized cohort as medical air rescue staff with this specific statement. The tremendous support for providing human factors training to all staff in medical facilities involved in patient care was notable in this survey and led to the recommendation that such trainings should be made available to all medical clinicians aiming to improve patient safety and staff satisfaction [29,34,35].

Another interesting aspect of this study was that over 90% of respondents confirmed that the skills they learned in the air rescue service were also applicable to and had a positive impact on their ground-based work. These results are in line with the recent literature [9] and may lead to the referral that CRM training approaches should also be made available to all ground-based rescue members by the rescue service organizations and other out-of-hospital agencies. Without doubt, further investigations with larger cohorts concerning the potential effects of human factors and appropriate trainings on ground-based paramedicine seem to be necessary.

For the dimension “Cooperation”, a consistent trend among responders was detected towards the agreement (54–95%) of items that direct to the benefits or a positive rating of cooperation or disagreement (62–70%) of items that are directed to a negative rating.

For the dimension “Safety”, the range of answers confirmed the observation from the current reality of care in Germany that staffing levels in hospitals are often insufficient to provide good care for all patients. From the response pattern of the questions on the dimension of “Safety”, one could deduce that there is a need for improvement with regard to patient safety in various German medical facilities, which has already been reported in earlier studies [7,36].

Finally, for this study, some limitations have to be mentioned. First of all, the results derived by the online survey are naturally based on the subjective judgement of the participants of the questionnaire and not by hard facts, such as less adverse events or less complications. As described above, the response rate was relatively low. A potential bias might be that the participants are employees of the service where the online survey was conducted, and it cannot be ruled out completely that they could be identified despite all anonymization efforts. Furthermore, it has to be noted that for some occupational groups, some statements of the modified FTPS only apply to a limited extent and could therefore only be answered to a limited extent. In addition, the occupational group of physicians who participated in the survey was significantly overrepresented, with 218 respondents compared with the HEMS TC with 35 participants. On the other hand, it can be seen as a great strength of the study that highly specialized and exclusive professionals participated in large numbers and that the results of the pilot study for the validation of the FTPS provided by Salem et al. (2008) could be confirmed in the current online survey.

## 5. Conclusions

In conclusion, the medical clinicians participating in this online survey believed that the skills they learned in human factors training such as CRM are helpful in their daily routine work in hospitals or other medical facilities, as well as in their ground-based rescue service activities. These findings may result in the recommendation to make CRM available to the medical staff in all medical facilities and also to ground-based rescue service staff on a regular basis aiming to increase patient safety and employee satisfaction.

## Figures and Tables

**Figure 1 medicines-10-00002-f001:**
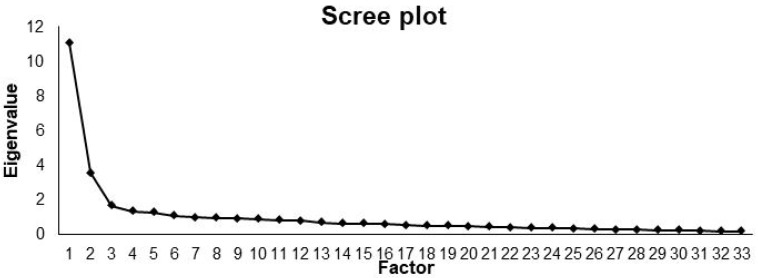
Eigenvalue progression as part of the factor analysis (Scree plot).

**Figure 2 medicines-10-00002-f002:**
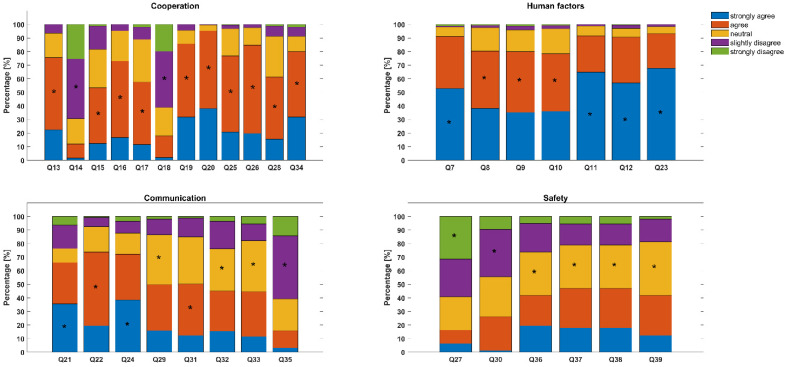
Results of Q13 to Q39 related to the specified factors. Asterisks within each stacked bar plot indicate the mode for ratings.

**Table 1 medicines-10-00002-t001:** Modified FTPS.

“TEAMWORK & PATIENT SAFETY” We ask you to fill out the questionnaire completely. The answer options are: - completely agree++ - agrees+ - neutral/ - rather does not agree -- - certainly does not agree - Please read each question (Q) carefully and then put a cross in the box that you think applies. Please tick the box that applies to you: Q1: Please check the box that applies to you: male __ female__ Q2: Please check the age box that applies to you: <35 years___/>35 years___ Q3: How long have you been in your professional medical practice outside of air ambulance service? <5 years___/5–10 years___/>5 years Q4: How long have you been involved in air rescue? <5 years___/5–10 years___/>5 years Q5: Please check the box that applies to you regarding your occupation in the hospital.						
Surgeon			Nurse Specialist Anesthesiology			
Anesthesiologist			Intensive care nurse specialist			
Internal physician			Other service group			
Physician in other specialty						
Q6: If you also work in ground-based rescue services, please tick the activity that applies to you.						
**Emergency Physician Ground-Based**						
Paramedic ground-based						
**Q**	**Questionnaire on Teamwork & Patient Safety**	**Agree Completely**	**Agree**	**Neutrally**	**Agree Rather Not**	**Agree Definitely Not**
7	The skills I learned in air rescue through CRM and similar courses, such as communication skills, situational awareness, and the like, can also be used in my daily clinical work, and they have a positive impact on what I do there.	++	+	/	-	--
8	By applying air rescue skills to my day-to-day activities in the clinic, I feel more confident and satisfied than I would without such a background.	++	+	/	-	--
9	By transferring skills and structures that I use and experience in air rescue, patient safety can be improved in my hospital.	++	+	/	-	--
10	By applying the skills acquired in air rescue, the cooperation between doctors, nurses and other professional groups can be improved in everyday hospital life.	++	+	/	-	--
11	Human Factors Training such as CRM and the skills that can be learned in it should be made available to all staff working in patient care at my hospital.	++	+	/	-	--
12	If you also work in ground-based rescue services: The skills learned in airborne rescue service through trainings like CRM can also be used in my ground-based job and have a positive effect on it.	++	+	/	-	--
13	In my hospital, comments and suggestions that come from nurses are considered.	++	+	/	-	--
14	It is difficult in my department to raise concerns about nursing or medical treatment issues.	++	+	/	-	--
15	Decisions are made with the involvement of all those affected.	++	+	/	-	--
16	Doctors, nurses and other professionals are a well-coordinated team here.	++	+	/	-	--
17	Disagreements are resolved appropriately (not “who is right” but “what is the best solution for the patient”).	++	+	/	-	--
18	I often cannot talk about discrepancies with the responsible doctors.	++	+	/	-	--
19	If you are not familiar with something, you can always ask questions about it.	++	+	/	-	--
20	I am supported by colleagues in the care/treatment of patients if I need help.	++	+	/	-	--
21	I know the first and last names of all the employees I was on duty with yesterday.	++	+	/	-	--
22	Important things are communicated reliably and understandably during the transfer of service.	++	+	/	-	--
23	Service handoff reports (to alert of potential hazards) are important for patient safety.	++	+	/	-	--
24	Meetings are held regularly at our hospital.	++	+	/	-	--
25	I am satisfied with the cooperation between me and the doctors of this hospital.	++	+	/	-	--
26	I am satisfied with the cooperation between me and the nursing staff of this hospital.	++	+	/	-	--
27	Our staffing levels are always sufficient to provide good care for all patients.	++	+	/	-	--
28	I would feel well and safe as a patient in this hospital.	++	+	/	-	--
29	Colleagues encourage me to report patient safety concerns.	++	+	/	-	--
30	Some employees more often disregard rules or guidelines that apply in their work area.	++	+	/	-	--
31	The atmosphere in this hospital helps individuals learn from the mistakes of others.	++	+	/	-	--
32	I get constructive feedback on my work performance.	++	+	/	-	--
33	Errors are dealt with appropriately in our hospital.	++	+	/	-	--
34	I would know who to contact with any concerns about patient safety.	++	+	/	-	--
35	It is difficult to talk about mistakes that have been made.	++	+	/	-	--
36	The management never knowingly compromises patient safety.	++	+	/	-	--
37	This hospital pays more attention to patient safety today than it did one year ago.	++	+	/	-	--
38	It is important to the management of the hospital that the highest attention is paid to patient safety.	++	+	/	-	--
39	Suggestions that would help improve patient safety would be implemented by the management.	++	+	/	-	--

**Table 2 medicines-10-00002-t002:** Rotated component matrix: representation of the factor loadings.

Loading	Factor			
Item (Q)	I. Cooperation	II. Human Factors	III. Communication	IV. Safety
Q13	0.83			
Q14	−0.635			
Q15	0.73		0.312	
Q16	0.694			
Q17	0.665			
Q18	−0.313			
Q19	0.518			
Q20	0.461			
Q25	0.499			
Q26	0.585			
Q28	0.481			
Q34	0.331			
Q7		0.82		
Q8		0.805		
Q9		0.814		
Q10		0.794		
Q11		0.623		
Q12		0.785		
Q23	0.106	0.141		
Q31	0.463		0.488	
Q21		0.302	0.500	
Q22			0.666	
Q24			0.326	
Q29	0.394		0.445	
Q32	0.303		0.579	
Q33	0.502		0.533	
Q35	−0.429		−0.451	
Q27			0.425	0.540
Q30				−0.307
Q36				0.584
Q37				0.794
Q38	0.309			0.758
Q39	0.385			0.604

**Table 3 medicines-10-00002-t003:** Results of items Q1 to Q6 (respondents’ general data) with row percentage in brackets.

Item	Number (Percentage)		
Q1 Gender	199 male (79%)	54 female (21%)	
Q2 Age	<35 years: 20 (8%)	>35 years: 233 (92%)	
Q3 How long have you been in your professional medical practice outside of air ambulance service?	<5 years: 2 (1%)	5–10 years: 30 (12%)	>5 years: 221 (87%)
Q4 How long have you been involved in air rescue?	<5 years: 83 (33%)	5–10 years: 71 (28%)	>5 years: 99 (39%)
Q5 Occupation in the hospital			
Surgeon	23 (9%)		
Anesthesiologist	185 (73%)		
Internal physician	6 (2%)		
Physician in other specialty	4 (1%)		
Nurse specialist Anesthesiology	16 (6%)		
Intensive care nurse specialist	19 (9%)		
Q6 If you also work in ground-based rescue services, please tick the activity that applies to you.			
Emergency physician ground-based	174 (75%)		
Paramedic ground-based	59 (25%)		

## Data Availability

The datasets used and/or analyzed during the current study are available from the corresponding author on reasonable request.

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
