# Peer review of "The Influence of Human Factors Training in Air Rescue Service on Patient Safety in Hospitals: Results of an Online Survey"

_medicines, 2022, doi:10.3390/medicines10010002_

Round 1
Reviewer 1 Report
I thank the Editor for giving me the opportunity to review this interesting paper by German colleagues. The paper is interesting and a it's good paper, especially because the methodology is straightforward, simple, clear and well executed. However, I have two minor concerns to bring to the Authors:
(a) Abstract
The abstract introduction needs to be revised; it is too specific, it does not give any background. In my opinion Authors need to remove all the initial questions and make a summary of the Introduction in the paper, so as to better contextualise the problem.
b) Questionnaire bias
There is an important data that the Authors stated also in the abstract, but did not adequately emphasised. Only 253 people out of 621 voluntarily took part in the questionnaire; this represents 40.7%. In addition to pointing out the percentage - which highlights well the share of interest in responding - the point is that this "plebiscite" in the answers comes from the staff who had the intention of answering the questionnaire and who therefore, in some way, found it interesting.
One could speculate as to why and what the people who decided not to participate in the questionnaire thought, but clearly this can be limited to discussion. The key point, however, is that the Authors have to point this out in the limits; moreover, in the Discussion and Conclusion, they have to be more cautious: instead of saying: "medical clinicians in this online survey believed that..." it is better to say: "medical clinicians that participated etc...".
Last but not least, the participants are employees of the service where they answered and can be identified. This bias has to be integrated into the limit considerations.
Otherwise, I confirm the goodness of the paper and hope to review the R1 again.
Author Response
Reviewer #1
I thank the Editor for giving me the opportunity to review this interesting paper by German colleagues. The paper is interesting and a it's good paper, especially because the methodology is straightforward, simple, clear and well executed. However, I have two minor concerns to bring to the Authors:
(a) Abstract
The abstract introduction needs to be revised; it is too specific, it does not give any background. In my opinion Authors need to remove all the initial questions and make a summary of the Introduction in the paper, so as to better contextualise the problem.
Answer: Dear Reviewer, we completely agree with you and have amended the abstract accordingly in order to clarify the background information also for interested readers who are not specialized in this topic.
- b) Questionnaire bias
There is an important data that the Authors stated also in the abstract, but did not adequately emphasised. Only 253 people out of 621 voluntarily took part in the questionnaire; this represents 40.7%. In addition to pointing out the percentage - which highlights well the share of interest in responding - the point is that this "plebiscite" in the answers comes from the staff who had the intention of answering the questionnaire and who therefore, in some way, found it interesting.
Answer: Thank you very much for this important question. We agree with you and have added the percentage to the abstract and to the M+M section. Additionally, we have dealt with the issue in the discussion in an additional paragraph.
One could speculate as to why and what the people who decided not to participate in the questionnaire thought, but clearly this can be limited to discussion. The key point, however, is that the Authors have to point this out in the limits.
Answer: Thank you for this valid question. You are completely right. We have amended this query in the limitiations as well as in the discussion and conclusion and also in the abstract accordingly.
Moreover, in the Discussion and Conclusion, they have to be more cautious: instead of saying: "medical clinicians in this online survey believed that..." it is better to say: "medical clinicians that participated etc...".
Answer: Thank you for this reference. We have renewed the wording in the conclusion accordingly.
Last but not least, the participants are employees of the service where they answered and can be identified. This bias has to be integrated into the limit considerations.
Answer: Thank you very much for this important question. We have added this bias to the limitations accordingly.
Otherwise, I confirm the goodness of the paper and hope to review the R1 again.
Answer: Thank you very much. We are very pleased that you like the manuscript.
Reviewer 2 Report
Major comments:
The introduction is sparse and is lacking sufficient background so that a naive reader could understand the work and its significance - work has been done, and referenced in this text but not described, on translating CRM from civil/military aviation to medical and hospital-based settings. A cursory review in the introduction is warranted.
I understand the interest in the identified questions presented starting L55, but placing the specific questions word for word here seem out of place, same thing for the abstract. Describing the general and specific interests would be more useful.
It is not clear to me which questions come from which surveys in this final adapted version. It would be informative to identify which questions remained the same and which were modified for the specific context of this study.
L121 your elbow or inflection point is at 3, you stated that content considerations led to 4 factors - it would be useful to describe in more detail this decision so others may replicate.
Table 3 - row percentages and measures of central tendency would be useful for interpretation of results.
L223 - a cursory online search produces multiple articles discussing CRM in ground medical transport situations...these should be reviewed and included
L229 and that homogeneous picture is - please explain
Did you review the personnel data on those who did not participate - as you indicate a majority were physicians - night it be useful to perform subgroup analyses on physicians, nurses. It may yield no difference or lead to only reporting physician respondents with the identified need to encourage nurse participation for future work
Minor comments:
L21 should be training, not trainings
L23 change "was" to "were" - same thing L65
L118 might you be referring to completed surveys, not data sets. Further did you analyze incomplete survey data?
Author Response
- The introduction is sparse and is lacking sufficient background so that a naive reader could understand the work and its significance - work has been done, and referenced in this text but not described, on translating CRM from civil/military aviation to medical and hospital-based settings. A cursory review in the introduction is warranted.
Answer: Dear reviewer, we completely agree with you and have amended the introduction accordingly including references in order to clarify the background information also for interested readers and to provide a cursory review of the current literature concerning CRM.
- I understand the interest in the identified questions presented starting L55, but placing the specific questions word for word here seem out of place, same thing for the abstract. Describing the general and specific interests would be more useful.
Answer: Dear reviewer, we have updated the abstract and the manuscript accordingly.
- It is not clear to me which questions come from which surveys in this final adapted version. It would be informative to identify which questions remained the same and which were modified for the specific context of this study.
Answer: Thank you for this important question. The original FTPS provided by Salem et al. 2008 (Q13 to Q39) was supplemented with Q1 to Q12 adapted to the specific context of the current study. We have added this information to the methods section accordingly.
- L121 your elbow or inflection point is at 3, you stated that content considerations led to 4 factors - it would be useful to describe in more detail this decision so others may replicate.
Answer: Thank you for your comment. The allocation of 4 factors was due on multiple criteria. Based on our factor analysis, 6 Items of the Questionnaire could not be assigned clearly to the first three factors but demonstrated a logical assignment to a fourth factor “patient safety”. Additionally, for selection of our Factors we also considered a cut-off eigenvalue of ≥ 1 (Kaiser criterion). In total this left 6 Factors, while the last two Factors in total only revealed 5 Items of the Questionnaire which could be also logically assigned to the first four factors. Therefore, to achieve logical data reduction with no overestimation of factors and therefore also smaller number of interpretable factors (with a larger number of logical items) a total of four factors were included in our analysis of the questionnaire. We added two new sections and a reference in the manuscript accordingly.
- Table 3 - row percentages and measures of central tendency would be useful for interpretation of results.
Answer: Thank you for your valuable comment. We now have added percentages in brackets for each value in Table 3. According central tendency, which - from our perspective - would only make sense for question items with a rating scale, we added the mode as an indicator of those ratings that were mentioned most frequently. We added the mode (indicated by asterisk) for each question in Figure 2. We also used colored bar plots to separate different ratings more clearly to the reader.
- L223 - a cursory online search produces multiple articles discussing CRM in ground medical transport situations...these should be reviewed and included.
Answer: Dear reviewer, thank you for this valuable query. We have addressed this topic and included a cursory overview to the introduction and discussion.
- L229 and that homogeneous picture is - please explain
Answer: Thank you for this comment. The terminus “homogeneous picture” was considered as that within the factor “cooperation“ that contained most items of the questionnaire, a very consistent trend was found among responders. In detail items that directed towards benefits or a positive rating of “cooperation“ were also positively confirmed (agree or strongly agree) by the responders while those items that directed towards to a negative rating were also consistently not confirmed by most responders. To avoid any confusion and to state this point more clearly we decided to rewrite this sentence in our manuscript.
- Did you review the personnel data on those who did not participate - as you indicate a majority were physicians - might it be useful to perform subgroup analyses on physicians, nurses. It may yield no difference or lead to only reporting physician respondents with the identified need to encourage nurse participation for future work.
Answer: Dear reviewer, we specifically examined only the personnel data of those who participated in the survey and not those who did not. This procedure is strictly based on the study by Salem et al. 2008 from the field of health services research, which we referred to in our study due to its scientific comprehensibility. The analysis provided by Salem et al., which was originally used to determine employee satisfaction in larger companies, primarily provides for an evaluation of the results of all service groups without subgroup analysis. This is also how we handled it. The potential reasons why more physicians than nurses participated in the survey have already been mentioned in the discussion. Nevertheless, a subgroup analysis may be the subject of future studies by our research group.
Minor comments:
- should be training, not trainings
Answer: Thank you for this valuable comment. The phrase has been amended accordingly in the text as well as in the headline.
- L23 change "was" to "were" - same thing L65
Answer: Thank you, the text has been rephrased accordingly.
- L118 might you be referring to completed surveys, not data sets. Further did you analyze incomplete survey data?
Answer: Dear reviewer, thank you for this valuable question. The wording has been amended accordingly. Only complete surveys have been analyzed in this study.
Round 2
Reviewer 2 Report
The authors adequately addressed each review point.